# Study on Springback Straightening after Bending of the U-Section of TC4 Material under High-Temperature Conditions

**DOI:** 10.3390/ma13081895

**Published:** 2020-04-17

**Authors:** Lu Jin, Yin-Fei Yang, Ren-Zheng Li, Ya-Wen Cui, Muhammad Jamil, Liang Li

**Affiliations:** 1College of Mechanical and Electrical Engineering, Nanjing University of Aeronautics and Astronautics, Nanjing 210016, China; jinl@nuaa.edu.cn (L.J.); engr.jamil@nuaa.edu.cn (M.J.); liliang@nuaa.edu.cn (L.L.); 2Chengdu Aircraft Industry Group Co., Ltd., Chengdu 610092, China; lrzazbb@126.com (R.-Z.L.); cuiyawen132@126.com (Y.-W.C.)

**Keywords:** TC4 material, U-section beam, bending deformation, springback straightening, stress relaxation

## Abstract

The thin-walled structures of titanium alloys have peculiar characteristics involving thin curved surfaces, complicated structures, and a poor rigidity. Therefore, bending or twisting distortion frequently occurs in forging, extrusion, drawing, transportation, cooling, and manufacturing. Straightening theory focuses on the straightening curvature or bending moment at room temperature, and a unified analytical model of the straightening curvature, the straightening bending moment, and the straightening stroke, as well as a study on springback straightening under high-temperature conditions, have not been investigated comprehensively. In order to understand the inherent mechanism of springback straightening and quantitative prediction of springback under high-temperature conditions, uniaxial tension tests were carried out to obtain the true stress–strain model of material and stress relaxation under the stress relaxation model. This paper is based on the theory of elastic-plastic mechanics and combines this with the mechanism of stress relaxation to establish springback and residual relative curvature equations of springback. The law of springback straightening is further explored, and springback and residual deflection equations are provided. The results of the study showed that the relative errors of the theoretical residual deflection of the bending deformation and residual deflections obtained by the experiment were less than 20%, with an average absolute error of less than 10%. Therefore, the hardening models adopted can achieve an allowable relative error if hardening parameters are properly selected. The proposed research provides basic data for the prediction of springback straightening, and the design of springback compensation tools can be applied in practical applications.

## 1. Introduction

Titanium alloys have peculiar characteristics, such as a high strength, temperature, and corrosion resistance. Therefore, titanium alloys are widely-adopted in aircraft engines, fuselages, and landing gears [1,2,3]. However, the fuselage used in the aerospace industry requires thin-walled structures, involving thin curved surfaces, complicated structures, and a poor rigidity. Distortion frequently occurs in forging, extrusion, drawing, transportation, cooling, and manufacturing due to external force and temperature changes. Furthermore, the length of thin-walled structures is generally more substantial, and when a large amount of metal must be removed in manufacturing, this can easily lead to bending or twisting distortion during the process. Failure to meet the design requirements and assembly accuracy severely affects the production efficiency and cost of the product. Therefore, the straightening process of bending distortion is required to improve workpiece straightness [4,5]. At present, reverse bending pressure straightening is a simple, flexible, economic, and effective method of straightening.

The straightening process of thin-walled structures is a typical elastic-plastic bending deformation process. The elastic-plastic mechanical properties of the workpiece material under the external force are directly related to the success of straightening. According to elastic-plastic mechanics, elastic-plastic deformation of the material is a complex non-linear problem, and the stress–strain relationship no longer follows Hook’s law. With the progress of the straightening process, the amount of plastic deformation continues to increase, the metal rheological strength also increases, and the yield limit may increase in one direction and decrease in the opposite direction. The stress and strain in the process are incredibly complicated. Since the prediction of this phenomenon and the design of springback compensation tools are still formidable challenges, many efforts have already been devoted to these issues.

Megharbel and Al-Qureshi [6,7] introduced an analytical analysis of the elastic-plastic bending of tubes and sections with different shapes. Analytical methods are given in the form of equations to provide a quantitative method for predicting the moment for forming the section of the tube based on a specific radius of curvature. Chakrabarty et al. [8,9] theoretically studied the elastic or plastic bending of sheet metal exhibiting a state of normal anisotropy. The results indicate that the elementary bending theory significantly overestimates the magnitude of the bending couple, producing elastic or plastic curvature of the bent sheet. Song et al. [10] studied the springback in T-section rails after lateral bending, considering the work-hardening materials. The authors established analytical formulas for the springback and residual curvatures. The numerical results indicate that material hardening directly affects the accuracy of springback prediction compared with experimental results. Johnson and Yu [11,12] determined the elastic springback in elastic work-hardening beams and plates under elastic-plastic pure bending and established a mathematical model for the final curvatures. Kosel et al. [13] investigated a repeated elastic-plastic pure plane bending/unbending process of beams made of a material with an elastic-linear hardening rheological model. Attention was devoted to beams having cross-sections with at least one axis of symmetry and that are initially straight or have a constant radius of curvature. Boris et al. [14] conducted a deflection analysis of beams with a rectangular cross-section under specific loading conditions, resulting in the greatest quadratic bending moment distribution and assuming elastic-plastic behavior with no hardening. Within the framework of the small strain and small displacement approach, analytical solutions are derived, which enable elastic-plastic analyses of beams to be performed in a closed systematic form. Li and Zhai [15,16] put forward a method for calculating the straightening stroke based on the straightening model. The straightening process model for shafts is developed according to the theory of elastic-plastic mechanics. By using the model, the straightening stroke can be calculated directly, according to the initial deflection of the bent part.

Regarding straightening theory, in contrast to the extensive literature describing the straightening curvature or bending moment at room temperature, little attention has been paid to establishing a unified analytical model of the straightening curvature, the straightening bending moment, and the straightening stroke, and studying springback straightening under high-temperature conditions.

In a nutshell, compared to cold straightening, the bending straightening of thin-walled structures of titanium alloys is usually heat treatment straightening, which has a different behavior at a high temperature. The straightening process becomes pretty complicated due to stress relaxation at a high temperature. This study establishes a unified analytical model of springback straightening to solve the problem of parameter transition from a theoretical model to practical application and obtains a stress relaxation model by relaxation experiments to solve the problem of complicated material behavior under a high temperature. In order to understand the inherent mechanism of springback straightening and to quantitatively predict the springback amount, this paper is based on the theory of elastic-plastic mechanics and combines it with the mechanism of stress relaxation to establish springback and residual relative curvature equations of springback at different bending degrees, respectively. The law of springback straightening is further explored, and springback and residual deflection equations are provided. Finally, the correctness of analytical models is presented, which was confirmed by the experiments.

## 2. Stress–Strain Curve

### 2.1. Experimental Setup and Specimen Preparation

The material of the beam was Ti6Al4V, and the material contents are listed in Table 1. According to GB/T 228.1-2010, the geometry of the specimen used for the uniaxial tension tests was as shown in Figure 1.

Three uniaxial tensile tests of the unprocessed specimen were carried out on the electro-hydraulic universal testing machine MTS, as shown in Figure 2. The tensile test specimen and the extensometer had to be installed, and the displacement rate of 0.2 mm/min was selected. After processing the measurement data according to Equations (1) and (2), the engineering stress–strain could be transformed into a true stress–strain curve, as shown in Figure 3.
(1)$$\sigma =\sigma_{nom}\left(1+m\right)$$
(2)$$\varepsilon =\ln\left(1+\varepsilon_{nom}\right)$$
where σ represents the true stress, ε represents the true strain $\sigma_{nom}$, describes the engineering stress, and $\varepsilon_{nom}$ represents the engineering strain.

Figure 3 shows the true stress-strain curve of TC4, which describes the stress–strain relationship of the material in uniaxial tensile tests. As can be seen from the above figure, the whole process can be clearly divided into two stages, namely the elastic deformation stage *OA* and the hardening stage *AB*. In the first stage, the stress–strain relationship is linear. It can be calculated that Young’s modulus *E* is 112 GPa and the elastic limit stress $\sigma_{e}$ is 850 MPa. When the strain further increases and exceeds the yield limit, the deformation of the material transitions into the second stage.

### 2.2. Material-Hardening Model

In order to accurately calculate the elastic-plastic springback of the U-section beam, it was combined with the true stress–strain curve obtained from the uniaxial tensile test in the previous section; the elastic-power hardening model was used to establish the constitutive relationship of the material to describe the true stress–strain relationship [13].
(3)$$\sigma =\{\begin{array}{cc} E\varepsilon & \varepsilon \leq \varepsilon_{s}\\ E\varepsilon -\eta {\left(\varepsilon -\varepsilon_{s}\right)}^{n} & \varepsilon >\varepsilon_{s} \end{array}$$
where η represents the material hardening coefficient, n represents the power strengthening coefficient, and n≥1.

Considering the different power hardening coefficients, the stress–strain curves of different elastic-power hardening models were obtained. The true stress–strain curves of the materials and the stress–strain curves of varying hardening coefficients were plotted together, as in Figure 4. It can be seen from the figure that the elastic-power hardening model is good agreement with the true stress–strain curve.

### 2.3. Stress Relaxation Model

The stress relaxation model shows that the phenomenon gradually decreases in internal stress with time, while the total strain remains constant, as shown in Figure 5. There are many factors affecting stress relaxation, including the temperature, time, initial stress, pre-deformation, and material properties [18,19,20].

It can be seen from Figure 5 that stress relaxation can be divided into two stages. The first stage includes the diffusion and viscous flow of grain boundaries. When the initial stress $\sigma_{0}$ is larger, the stress decreases faster, but $\sigma_{0}$ has little effect on the stress relaxation rate of the second stage. The rate of the stress decrease in the first stage is higher than that in the second stage, and the final stress decreases to a specific equilibrium value of $\sigma_{r}$. When the temperature is high enough, $\sigma_{r}$ will approach zero. Some studies have shown that the temperature has a significant effect on stress relaxation. The higher the temperature is, the more significant the stress reduction is; the longer the time is, the higher the stress reduction is, but the effect of temperature on stress relaxation is significantly greater than that of time.

When heat treatment is used to correct the shape, because the part is pressed in the mold, it can be considered a stress relaxation process with constant strain. During heat treatment, the workpiece will undergo stress relaxation, thereby eliminating the elastic stress. Introducing the stress relaxation model into the equation can lay the foundation for the establishment of the high-temperature springback curvature equation. According to Figure 5, it can be seen that the stress relaxation equation changes according to the exponential law or the logarithmic law, so this paper uses the natural logarithmic equation to describe the relaxation law, as follows:(4)$$\sigma =\sigma_{0}-S\cdot \left(1+V\cdot T\right)$$
where σ represents instantaneous stress, $\sigma_{0}$ represents the initial stress, T represents time, and *S* and V represent stress relaxation coefficients.

The experimental specimens were 0.8 mm thick TC4 plates with annealing temperatures of 650, 700, and 750 °C, respectively. The stress relaxation experiments were all performed on a JY-I type high-temperature stress relaxation machine, equipped with a closed-loop control system. This provided constant displacement for stress relaxation experiments, and the moving speed of the drawing rod was set to 6 mm/min, as shown in Figure 6.

The stress relaxation test data was processed by mathematical statistics methods to obtain the stress relaxation equation, as shown in Table 2.

It can be seen from Figure 7 that the stress relaxation law of TC4 material at a high temperature shows consistency with the typical stress relaxation law of Figure 5. At the same temperature, when $\sigma_{0}$ is larger, the stress relaxation decreases faster in the first stage; when $\sigma_{0}$ is the same, the decrease rate of stress relaxation increases with an increasing temperature, especially for the first stage of the stress relaxation rate, at which point the impact is the most significant.

## 3. Development of Regression Models for Springback

### 3.1. Assumptions about Bending Deformation

During the bending deformation process, the workpiece has undergone two interrelated deformation stages of elasticity and plasticity, and the cross-section of the workpiece has necessarily undergone a gradual deformation process from the outer surface to the inside. In order to facilitate the analytical analysis, it was assumed that the workpiece material was uniform and isotropic, and followed the Euler–Bernoulli beam theory [21]:

(1) Plane assumption: The cross-section of the workpiece after bending is still flat, and the shape and size of the cross-section remain constant before and after bending;

(2) Unidirectional stress assumption: During bending deformation, there is no extrusion between the longitudinal fibers of the workpiece, and there is no transverse stress. The stress–strain relationship of the fiber tensile and compression is the same, and it is always in the unidirectional stress state;

(3) It is assumed that the geometric neutral axis, the stress neutral axis, and the strain neutral axis of the workpiece coincide.

### 3.2. Theoretical, Analytical Equations of Bending and Springback

The U-section beam size and the stress–strain relationship under different bending degrees are shown in Figure 8. During the bending moment application, the workpiece first undergoes elastic deformation; then yields a point, and finally, the web transitions into the plastic stage. The whole process can be divided into three stages according to the degree of plastic deformation: (1) plastic deformation occurs at the upper end, and the stress–strain relationship is shown in Figure 7b; (2) plastic deformation penetrates the bottom web, and the stress–strain relationship is shown Figure 7c; (3) all webs undergo plastic deformation, and the stress–strain relationship is shown in Figure 7d.

#### 3.2.1. Plastic Region Occurrence at the Top End

When the plastic region occurs at the upper end, the elastic region range is $h_{1}<h_{a}<h_{2}$, and its stress–strain relationship is shown in Figure 7. According to the balance moment rule, the expression of the bending moment can be expressed as
(5)$$M_{\lambda 1}=2\int_{h_{a}}^{h_{2}}a\left[\sigma_{s}\frac{y}{h_{a}}-\eta {\left(\frac{y}{h_{a}}-1\right)}^{n}\right]ydy+2\int_{0}^{h_{a}}a\sigma_{s}\frac{y}{h_{a}}ydy+2\int_{0}^{h_{1}-t}a\sigma_{s}\frac{y}{h_{a}}ydy+\int_{h_{1}-t}^{h_{1}}B\sigma_{s}\frac{y}{h_{a}}ydy$$
where $\sigma_{s}$ represents the yield limit of the material and *σ* expresses the fiber layer stress of the elastic from the neutral layer y.

When the outermost fiber reaches the yield limit of the material, that is, $h_{a}=h_{2}$, the maximum limit bending moment *M_a_* can be expressed as
(6)$$M_{a}=2\int_{0}^{h_{1}-t}a\sigma_{s}\frac{y}{h_{a}}ydy+\int_{h_{1}-t}^{h_{1}}B\sigma_{s}\frac{y}{h_{a}}ydy+2\int_{0}^{h_{2}}a\sigma_{s}\frac{y}{h_{a}}ydy$$

The elastic limit bending moment expression of the U-section can then be obtained. It can be expressed as
(7)$$M_{a}=\frac{Bh^{3}\sigma_{s}}{3h_{2}}\left[f_{1}\left(G,M,D\right)+f_{2}\left(G,M\right)\right]$$
where $D=t/h;\;G=a/B;\;M=h_{1}/h;\;f_{1}\left(G,M,D\right)=\left(1-2G\right){\left(D-M\right)}^{3};\;f_{2}\left(G,M\right)=M^{3}+2G{\left(1-M\right)}^{3}.$

$M_{\lambda 1}^{\prime }=M/M_{a}$ can be defined as the bending moment ratio, then
(8)$$M_{\lambda 1}^{\prime }=-\frac{6\eta h_{a}^{2}h_{2}}{Bh^{3}\sigma_{s}\left(f_{1}+f_{2}\right)}\left[\frac{{\left(\frac{h_{2}}{h_{a}}-1\right)}^{n+2}}{n+2}+\frac{{\left(\frac{h_{2}}{h_{a}}-1\right)}^{n+1}}{n+1}\right]+\frac{2Gh_{2}}{h^{3}\left(f_{1}+f_{2}\right)h_{a}}\left[h_{2}^{3}+{\left(h_{1}-t\right)}^{3}\right]\;+\frac{h_{2}}{h^{3}\left(f_{1}+f_{2}\right)h_{a}}\left[h_{1}^{3}-{\left(h_{1}-t\right)}^{3}\right]$$

At high temperatures, due to stress relaxation, the hardening coefficient of the material η changes with time, but the external force $\sigma_{0}$ determines η. At the same time, the outer fiber strain $y=h_{2}$ and the corresponding stress and stress relaxation Equation (4) can be substituted into the Equation (8), and
(9)$$M_{\lambda 1}^{\prime }=-\frac{6ah_{2}^{3}\xi^{2}}{Bh^{3}\sigma_{s}\left(f_{1}+f_{2}\right)}\left[\frac{\sigma_{s}\xi^{-1}-\sigma_{0}+S\cdot \ln\left(1+V\cdot T\right)}{{\left(\xi^{-1}-1\right)}^{n}}\right]\left[\frac{{\left(\xi^{-1}-1\right)}^{n+2}}{\left(n+2\right)}+\frac{{\left(\xi^{-1}-1\right)}^{n+1}}{\left(n+1\right)}\right]\;+\frac{2G\xi^{-1}}{h^{3}\left(f_{1}+f_{2}\right)}\left[h_{2}^{3}+{\left(h_{1}-t\right)}^{3}\right]+\frac{\xi^{-1}}{h^{3}\left(f_{1}+f_{2}\right)}\left[h_{1}^{3}-{\left(h_{1}-t\right)}^{3}\right]$$
where $\xi =h_{a}/h_{2}$ represents the elastic region ratio and $h_{1}/h_{2}\leq \xi \leq 1$.

$K_{\sum }^{\prime }$ is the relative curvature of the U-section beam, and $K_{f}^{\prime }$ is the relative curvature of the springback. Based on the elastic-plastic bending and elastic mechanics plane assumption,
(10)$$\xi =\frac{1}{K_{\sum }^{\prime }}$$
(11)$$M^{\prime }=K_{f}^{\prime }$$

By substituting Equation (9) into Equations (10) and (11), $K_{f}^{\prime }$ can be expressed as
(12)$$K_{f}^{\prime }=-\frac{6ah_{2}^{3}K_{\sum }^{\prime -2}}{Bh^{3}\sigma_{s}\left(f_{1}+f_{2}\right)}\left[\frac{\sigma_{s}K_{\sum }^{\prime }-\sigma_{0}+S\cdot \ln\left(1+V\cdot T\right)}{{\left(K_{\sum }^{\prime }-1\right)}^{n}}\right]\left[\frac{{\left(K_{\sum }^{\prime }-1\right)}^{n+2}}{\left(n+2\right)}+\frac{{\left(K_{\sum }^{\prime }-1\right)}^{n+1}}{\left(n+1\right)}\right]\;+\frac{2GK_{\sum }^{\prime }}{h^{3}\left(f_{1}+f_{2}\right)}\left[h_{2}^{3}+{\left(h_{1}-t\right)}^{3}\right]+\frac{K_{\sum }^{\prime }}{h^{3}\left(f_{1}+f_{2}\right)}\left[h_{1}^{3}-{\left(h_{1}-t\right)}^{3}\right]$$

After springback, the relative residual curvature $K_{c}^{\prime }$ is
(13)$$K_{c}^{\prime }=K^{\prime }-K_{f}^{\prime }$$

If the relative initial curvature $K_{0}^{\prime }$ is equal to zero, then the relative total curvature $K_{\sum }^{\prime }$ can be expressed as
(14)$$K_{\sum }^{\prime }=K^{\prime }+K_{0}^{\prime }=K^{\prime }$$

By substituting Equation (12) into Equations (13) and (14), $K_{c}^{\prime }$ can be expressed as
(15)$$K_{c}^{\prime }=K_{\sum }^{\prime }+\frac{6ah_{2}^{3}K_{\sum }^{\prime -2}}{Bh^{3}\sigma_{s}\left(f_{1}+f_{2}\right)}\left[\frac{\sigma_{s}K_{\sum }^{\prime }-\sigma_{0}+S\cdot \ln\left(1+V\cdot T\right)}{{\left(K_{\sum }^{\prime }-1\right)}^{n}}\right]\left[\frac{{\left(K_{\sum }^{\prime }-1\right)}^{n+2}}{\left(n+2\right)}+\frac{{\left(K_{\sum }^{\prime }-1\right)}^{n+1}}{\left(n+1\right)}\right]\;-\frac{2GK_{\sum }^{\prime }}{h^{3}\left(f_{1}+f_{2}\right)}\left[h_{2}^{3}+{\left(h_{1}-t\right)}^{3}\right]-\frac{K_{\sum }^{\prime }}{h^{3}\left(f_{1}+f_{2}\right)}\left[h_{1}^{3}-{\left(h_{1}-t\right)}^{3}\right]$$

When $M_{\lambda 1}^{\prime }$ is given, $K_{\sum }^{\prime }$, $K_{f}^{\prime }$ and $K_{c}^{\prime }$ can be obtained from Equations (12) and (15).

#### 3.2.2. Plastic Region in the Bottom Web

When the plastic region penetrates into the bottom web, the elastic region range is $2\left(h_{1}-t\right)<h_{a}<2h_{1}$, and its stress–strain relationship is as shown in Figure 7. Therefore, the expression of the bending moment can be expressed as
(16)$$M_{\lambda 2}=2\int_{\frac{h_{a}}{2}}^{h_{2}}a\left[\sigma_{s}\frac{2y}{h_{a}}-\eta {\left(\frac{2y}{h_{a}}-1\right)}^{n}\right]ydy+2\int_{\frac{h_{a}}{2}}^{h_{1}}B\left[\sigma_{s}\frac{2y}{h_{a}}-\eta {\left(\frac{2y}{h_{a}}-1\right)}^{n}\right]ydy\;+2\int_{0}^{\frac{h_{a}}{2}}a\sigma ydy+2\int_{0}^{h_{1}-t}a\sigma ydy+\int_{h_{1}-t}^{\frac{h_{a}}{2}}B\sigma ydy$$
where $\sigma_{s}$ represents the yield limit of the material and σ represents the fiber layer stress of the elastic from the neutral layer y.

Furthermore, combined with stress relaxation,
(17)$$M_{\lambda 2}^{\prime }=-\frac{6ah_{2}^{3}\xi^{2}}{Bh^{3}\sigma_{s}\left(f_{1}+f_{2}\right)}\left[\frac{\sigma_{s}\xi^{-1}-\sigma_{0}+S\cdot \ln\left(1+V\cdot T\right)}{{\left(\xi^{-1}-1\right)}^{n}}\right]\left[\frac{{\left(\xi^{-1}-1\right)}^{n+2}}{\left(n+2\right)}+\frac{{\left(\xi^{-1}-1\right)}^{n+1}}{\left(n+1\right)}\right]\;-\frac{3ah_{2}^{3}\xi^{2}}{Bh^{3}\sigma_{s}\left(f_{1}+f_{2}\right)}\left[\frac{\sigma_{s}\xi^{-1}-\sigma_{0}+S\cdot \ln\left(1+V\cdot T\right)}{{\left(\xi^{-1}-1\right)}^{n}}\right]\left[\frac{{\left(\frac{h_{1}}{h_{2}}\xi^{-1}-1\right)}^{n+2}}{\left(n+2\right)}+\frac{{\left(\frac{h_{1}}{h_{2}}\xi^{-1}-1\right)}^{n+1}}{\left(n+1\right)}\right]\;+\frac{2G\xi^{-1}}{h^{3}\left(f_{1}+f_{2}\right)}\left[h_{2}^{3}+{\left(h_{1}-t\right)}^{3}\right]+\frac{\xi^{-1}}{h^{3}\left(f_{1}+f_{2}\right)}\left[h_{1}^{3}-{\left(h_{1}-t\right)}^{3}\right]$$
where $\xi =h_{a}/2h_{2}$ represents the elastic region ratio, and $\left(h_{1}-t\right)/h_{2}\leq \xi \leq h_{1}/h_{2}$.

By substituting Equation (17) into Equations (10) and (11), the relationship $K_{f}^{\prime }$ can be expressed as
(18)$$K_{f}^{\prime }=-\frac{6ah_{2}^{3}K_{\sum }^{\prime -2}}{Bh^{3}\sigma_{s}\left(f_{1}+f_{2}\right)}\left[\frac{\sigma_{s}K_{\sum }^{\prime }-\sigma_{0}+S\cdot \ln\left(1+V\cdot T\right)}{{\left(K_{\sum }^{\prime }-1\right)}^{n}}\right]\left[\frac{{\left(K_{\sum }^{\prime }-1\right)}^{n+2}}{\left(n+2\right)}+\frac{{\left(K_{\sum }^{\prime }-1\right)}^{n+1}}{\left(n+1\right)}\right]\;-\frac{3ah_{2}^{3}K_{\sum }^{\prime -2}}{Bh^{3}\sigma_{s}\left(f_{1}+f_{2}\right)}\left[\frac{\sigma_{s}K_{\sum }^{\prime }-\sigma_{0}+S\cdot \ln\left(1+V\cdot T\right)}{{\left(K_{\sum }^{\prime }-1\right)}^{n}}\right]\left[\frac{{\left(\frac{h_{1}}{h_{2}}K_{\sum }^{\prime }-1\right)}^{n+2}}{\left(n+2\right)}+\frac{{\left(\frac{h_{1}}{h_{2}}K_{\sum }^{\prime }-1\right)}^{n+1}}{\left(n+1\right)}\right]\;+\frac{2GK_{\sum }^{\prime }}{h^{3}\left(f_{1}+f_{2}\right)}\left[h_{2}^{3}+{\left(h_{1}-t\right)}^{3}\right]+\frac{K_{\sum }^{\prime }}{h^{3}\left(f_{1}+f_{2}\right)}\left[h_{1}^{3}-{\left(h_{1}-t\right)}^{3}\right]$$

By substituting Equation (18) into Equations (13) and (14), the relationship $K_{c}^{\prime }$ can be expressed as
(19)$$K_{c}^{\prime }=K_{\sum }^{\prime }+\frac{6ah_{2}^{3}K_{\sum }^{\prime -2}}{Bh^{3}\sigma_{s}\left(f_{1}+f_{2}\right)}\left[\frac{\sigma_{s}K_{\sum }^{\prime }-\sigma_{0}+S\cdot \ln\left(1+V\cdot T\right)}{{\left(K_{\sum }^{\prime }-1\right)}^{n}}\right]\left[\frac{{\left(K_{\sum }^{\prime }-1\right)}^{n+2}}{\left(n+2\right)}+\frac{{\left(K_{\sum }^{\prime }-1\right)}^{n+1}}{\left(n+1\right)}\right]\;+\frac{3ah_{2}^{3}K_{\sum }^{\prime -2}}{Bh^{3}\sigma_{s}\left(f_{1}+f_{2}\right)}\left[\frac{\sigma_{s}K_{\sum }^{\prime }-\sigma_{0}+S\cdot \ln\left(1+V\cdot T\right)}{{\left(K_{\sum }^{\prime }-1\right)}^{n}}\right]\left[\frac{{\left(\frac{h_{1}}{h_{2}}K_{\sum }^{\prime }-1\right)}^{n+2}}{\left(n+2\right)}+\frac{{\left(\frac{h_{1}}{h_{2}}K_{\sum }^{\prime }-1\right)}^{n+1}}{\left(n+1\right)}\right]\;-\frac{2GK_{\sum }^{\prime }}{h^{3}\left(f_{1}+f_{2}\right)}\left[h_{2}^{3}+{\left(h_{1}-t\right)}^{3}\right]-\frac{K_{\sum }^{\prime }}{h^{3}\left(f_{1}+f_{2}\right)}\left[h_{1}^{3}-{\left(h_{1}-t\right)}^{3}\right]$$
When $M_{\lambda 2}^{\prime }$ is given, $K_{\sum }^{\prime }$, $K_{f}^{\prime }$ and $K_{c}^{\prime }$ can be obtained from Equation (18) and Equation (19).

#### 3.2.3. All Webs Are in a Plastic State

When all the webs are in a plastic state, the elastic region range is $0\leq h_{a}\leq 2\left(h_{1}-t\right)$, and the stress–strain relationship is as shown in Figure 7. Then, the expression of the bending moment can be expressed as
(20)$$M_{\lambda 3}=4\int_{0}^{\frac{h_{a}}{2}}a\sigma ydy+2\int_{\frac{h_{a}}{2}}^{h_{2}}a\left[\sigma_{s}\frac{2y}{h_{a}}-\eta {\left(\frac{2y}{h_{a}}-1\right)}^{n}\right]ydy+2\int_{\frac{h_{a}}{2}}^{h_{1}-t}a\left[\sigma_{s}\frac{2y}{h_{a}}-\eta {\left(\frac{2y}{h_{a}}-1\right)}^{n}\right]ydy\;+2\int_{h_{1}-t}^{h_{1}}B\left[\sigma_{s}\frac{2y}{h_{a}}-\eta {\left(\frac{2y}{h_{a}}-1\right)}^{n}\right]ydy$$

Furthermore, combined with stress relaxation,
(21)$$M_{\lambda 3}^{\prime }=-\frac{6ah_{2}^{3}\xi^{2}}{Bh^{3}\sigma_{s}\left(f_{1}+f_{2}\right)}\left[\frac{\sigma_{s}\xi^{-1}-\sigma_{0}+S\cdot \ln\left(1+V\cdot T\right)}{{\left(\xi^{-1}-1\right)}^{n}}\right]\left[\frac{{\left(\xi^{-1}-1\right)}^{n+2}}{\left(n+2\right)}+\frac{{\left(\xi^{-1}-1\right)}^{n+1}}{\left(n+1\right)}\right]\;-\frac{6ah_{2}^{3}\xi^{2}}{Bh^{3}\sigma_{s}\left(f_{1}+f_{2}\right)}\left[\frac{\sigma_{s}\xi^{-1}-\sigma_{0}+S\cdot \ln\left(1+V\cdot T\right)}{{\left(\xi^{-1}-1\right)}^{n}}\right]\left[\frac{{\left(\frac{h_{1}-t}{h_{2}}\xi^{-1}-1\right)}^{n+2}}{\left(n+2\right)}+\frac{{\left(\frac{h_{1}-t}{h_{2}}\xi^{-1}-1\right)}^{n+1}}{\left(n+1\right)}\right]\;-\frac{3ah_{2}^{3}\xi^{2}}{Bh^{3}\sigma_{s}\left(f_{1}+f_{2}\right)}\left[\frac{\sigma_{s}\xi^{-1}-\sigma_{0}+S\cdot \ln\left(1+V\cdot T\right)}{{\left(\xi^{-1}-1\right)}^{n}}\right]\;\cdot \left[\frac{{\left(\frac{h_{1}}{h_{2}}\xi^{-1}-1\right)}^{n+2}}{\left(n+2\right)}+\frac{{\left(\frac{h_{1}}{h_{2}}\xi^{-1}-1\right)}^{n+1}}{\left(n+1\right)}-\frac{{\left(\frac{h_{1}-t}{h_{2}}\xi^{-1}-1\right)}^{n+2}}{\left(n+2\right)}-\frac{{\left(\frac{h_{1}-t}{h_{2}}\xi^{-1}-1\right)}^{n+1}}{\left(n+1\right)}\right]\;+\frac{4ah_{2}^{3}\xi^{2}}{Bh^{3}\left(f_{1}+f_{2}\right)}+\frac{4ah_{2}^{3}\xi^{-1}}{Bh^{3}\left(f_{1}+f_{2}\right)}\left[\frac{{\left(h_{1}-t\right)}^{3}}{2h_{2}^{3}}-\xi^{3}+\frac{1}{2}\right]+\frac{\xi^{-1}}{h^{3}\left(f_{1}+f_{2}\right)}\left[h_{1}^{3}-{\left(h_{1}-t\right)}^{3}\right]$$
where $\xi =h_{a}/2h_{2}$ represents the elastic region ratio, and $0\leq \xi \leq \left(h_{1}-t\right)/h_{2}$.

Substituting Equation (21) into Equation (10) and Equation (11), the relationship $K_{f}^{\prime }$ can be expressed as
(22)$$K_{f}^{\prime }=-\frac{6ah_{2}^{3}K_{\sum }^{\prime -2}}{Bh^{3}\sigma_{s}\left(f_{1}+f_{2}\right)}\left[\frac{\sigma_{s}K_{\sum }^{\prime }-\sigma_{0}+S\cdot \ln\left(1+V\cdot T\right)}{{\left(K_{\sum }^{\prime }-1\right)}^{n}}\right]\left[\frac{{\left(K_{\sum }^{\prime }-1\right)}^{n+2}}{\left(n+2\right)}+\frac{{\left(K_{\sum }^{\prime }-1\right)}^{n+1}}{\left(n+1\right)}\right]\;-\frac{6ah_{2}^{3}K_{\sum }^{\prime -2}}{Bh^{3}\sigma_{s}\left(f_{1}+f_{2}\right)}\left[\frac{\sigma_{s}K_{\sum }^{\prime }-\sigma_{0}+S\cdot \ln\left(1+V\cdot T\right)}{{\left(K_{\sum }^{\prime }-1\right)}^{n}}\right]\left[\frac{{\left(\frac{h_{1}-t}{h_{2}}K_{\sum }^{\prime }-1\right)}^{n+2}}{\left(n+2\right)}+\frac{{\left(\frac{h_{1}-t}{h_{2}}K_{\sum }^{\prime }-1\right)}^{n+1}}{\left(n+1\right)}\right]\;-\frac{3ah_{2}^{3}K_{\sum }^{\prime -2}}{Bh^{3}\sigma_{s}\left(f_{1}+f_{2}\right)}\left[\frac{\sigma_{s}K_{\sum }^{\prime }-\sigma_{0}+S\cdot \ln\left(1+V\cdot T\right)}{{\left(K_{\sum }^{\prime }-1\right)}^{n}}\right]\;\cdot \left[\frac{{\left(\frac{h_{1}}{h_{2}}K_{\sum }^{\prime }-1\right)}^{n+2}}{\left(n+2\right)}+\frac{{\left(\frac{h_{1}}{h_{2}}K_{\sum }^{\prime }-1\right)}^{n+1}}{\left(n+1\right)}-\frac{{\left(\frac{h_{1}-t}{h_{2}}K_{\sum }^{\prime }-1\right)}^{n+2}}{\left(n+2\right)}-\frac{{\left(\frac{h_{1}-t}{h_{2}}K_{\sum }^{\prime }-1\right)}^{n+1}}{\left(n+1\right)}\right]\;+\frac{4ah_{2}^{3}K_{\sum }^{\prime -2}}{Bh^{3}\left(f_{1}+f_{2}\right)}+\frac{4ah_{2}^{3}K_{\sum }^{\prime }}{Bh^{3}\left(f_{1}+f_{2}\right)}\left[\frac{{\left(h_{1}-t\right)}^{3}}{2h_{2}^{3}}-K_{\sum }^{\prime -3}+\frac{1}{2}\right]+\frac{K_{\sum }^{\prime }}{h^{3}\left(f_{1}+f_{2}\right)}\left[h_{1}^{3}-{\left(h_{1}-t\right)}^{3}\right]$$

By substituting Equation (22) into Equations (13) and (14), the relationship of $K_{c}^{\prime }$ can be expressed as
(23)$$K_{c}^{\prime }=K_{\sum }^{\prime }+\frac{6ah_{2}^{3}K_{\sum }^{\prime -2}}{Bh^{3}\sigma_{s}\left(f_{1}+f_{2}\right)}\left[\frac{\sigma_{s}K_{\sum }^{\prime }-\sigma_{0}+S\cdot \ln\left(1+V\cdot T\right)}{{\left(K_{\sum }^{\prime }-1\right)}^{n}}\right]\left[\frac{{\left(K_{\sum }^{\prime }-1\right)}^{n+2}}{\left(n+2\right)}+\frac{{\left(K_{\sum }^{\prime }-1\right)}^{n+1}}{\left(n+1\right)}\right]\;+\frac{6ah_{2}^{3}K_{\sum }^{\prime -2}}{Bh^{3}\sigma_{s}\left(f_{1}+f_{2}\right)}\left[\frac{\sigma_{s}K_{\sum }^{\prime }-\sigma_{0}+S\cdot \ln\left(1+V\cdot T\right)}{{\left(K_{\sum }^{\prime }-1\right)}^{n}}\right]\left[\frac{{\left(\frac{h_{1}-t}{h_{2}}K_{\sum }^{\prime }-1\right)}^{n+2}}{\left(n+2\right)}+\frac{{\left(\frac{h_{1}-t}{h_{2}}K_{\sum }^{\prime }-1\right)}^{n+1}}{\left(n+1\right)}\right]\;+\frac{3ah_{2}^{3}K_{\sum }^{\prime -2}}{Bh^{3}\sigma_{s}\left(f_{1}+f_{2}\right)}\left[\frac{\sigma_{s}K_{\sum }^{\prime }-\sigma_{0}+S\cdot \ln\left(1+V\cdot T\right)}{{\left(K_{\sum }^{\prime }-1\right)}^{n}}\right]\;\cdot \left[\frac{{\left(\frac{h_{1}}{h_{2}}K_{\sum }^{\prime }-1\right)}^{n+2}}{\left(n+2\right)}+\frac{{\left(\frac{h_{1}}{h_{2}}K_{\sum }^{\prime }-1\right)}^{n+1}}{\left(n+1\right)}-\frac{{\left(\frac{h_{1}-t}{h_{2}}K_{\sum }^{\prime }-1\right)}^{n+2}}{\left(n+2\right)}-\frac{{\left(\frac{h_{1}-t}{h_{2}}K_{\sum }^{\prime }-1\right)}^{n+1}}{\left(n+1\right)}\right]\;-\frac{4ah_{2}^{3}K_{\sum }^{\prime -2}}{Bh^{3}\left(f_{1}+f_{2}\right)}+\frac{4ah_{2}^{3}K_{\sum }^{\prime }}{Bh^{3}\left(f_{1}+f_{2}\right)}\left[\frac{{\left(h_{1}-t\right)}^{3}}{2h_{2}^{3}}-K_{\sum }^{\prime -3}+\frac{1}{2}\right]-\frac{K_{\sum }^{\prime }}{h^{3}\left(f_{1}+f_{2}\right)}\left[h_{1}^{3}-{\left(h_{1}-t\right)}^{3}\right]$$
When $M_{\lambda 3}^{\prime }$ is given, $K_{\sum }^{\prime }$, $K_{f}^{\prime }$ and $K_{c}^{\prime }$ can be obtained from Equations (22) and (23).

## 4. Results and Discussion

To facilitate data processing and research, the initial curvature of the workpiece was set to zero, that is, $K_{0}=0$. According to the material mechanics formula, the symmetrical support at both ends could be obtained, and the bending midpoint of the workpiece was straightened. After unloading, the bent workpiece could be measured to get the rebound and residual deflection of the workpiece, which can be expressed as
(24)$$\varphi_{f}=\varphi_{\sum }-\varphi_{c}$$
where $\varphi_{\sum }$ represents the loading stroke, $\varphi_{f}$ represents the elastic deflection value, and $\varphi_{c}$ represents the residual deflection value.

The theoretical relationship between the workpiece loading stroke, elastic deflection, and residual deflection could be obtained using the following formula:(25)$$\varphi_{f}=\frac{\lambda M_{a}{\left(2l\right)}^{2}}{12EI}M^{\prime }=\lambda \varphi_{a}M^{\prime }=\lambda \varphi_{a}K_{f}^{\prime }$$
(26)$$\varphi_{\sum }=\varphi_{f}+\varphi_{c}=\lambda \varphi_{a}K^{\prime }$$

The material of the U-section beam was TC4. The material properties are shown in Table 3. Under the condition of 650 °C, λ=1, n=1.01, and the total beam length was 2*l*. Combined with the stress relaxation equation of $\sigma_{0}=30.6$ MPa and substituting them into the high-temperature springback analytical equation of the previous section, the relative curvature equation could be obtained and then converted into the stress relaxation at 1 minute, 2 minutes, and 5 minutes. Equations (24) to (26) could then be used to convert these values into the elastic and residual deflection equations.

It can be seen from Figure 9 that with the increase of the relative curvature of the bend, the relative curvature of the springback gradually increases, but the rate gradually decreases. In this paper, the constitutive relationship of the material is established based on the elastic-power hardening model. The stress relaxation time under a high temperature is different, and the change law of the relative curvature of springback is the same. When the bending relative curvature is constant, the stress relaxation time is longer and the relative springback curvature is smaller.

Regarding Figure 5, it can be seen that due to the lattice distortion caused by elastic deformation, the dislocations and cavity structures generated by plastic deformation have high potential energy. With the increase of the high-temperature time, this transformation will be strengthened, and the elastic deformation will gradually become unrecovered creep deformation, representing stress relaxation. The corresponding elastic stress becomes smaller and smaller, the elastic strain part of the total strain is reduced, and the plastic strain part is increased, so that the stress is gradually reduced and the relative springback curvature is reduced. To verify the accuracy of the theoretical model, different holding times and loading strokes were adopted to obtain a series of springback and residual deflection values by the high-temperature stress relaxation machine, as shown in Figure 6, and experimental data are shown in Table 4.

In order to compare and analyze the springback law of the U-section beam under high-temperature conditions, the curve of the residual deflection of the theoretical model and experiments was plotted and is shown in Figure 10.

It can be seen from Figure 10 that the springback and residual deflection equations are consistent with the change law of their relative curvature equations. Using the formulas of Equations (24)–(26), they were converted into the relationship formula, which more intuitively describes the relationship between the physical quantities and lays the foundation for subsequent verification work. From Table 3 and Figure 10, it can be seen that the relative errors of the theoretical residual deflection of the bending deformation and the residual deflections obtained by the experiment are less than 20%, with an average absolute error of less than 10%. Therefore, although there is an apparent yield stage in the true stress–strain curve, the adopted hardening models can achieve an allowable relative error, if hardening parameters are appropriately selected.

By comparing and analyzing the theoretical model and experiment, it can be seen that in the initial stage of bending deformation, the error between the experimental result and the theoretical model calculation value is large. At the same time, as the bending deformation increases, the value of the experimental result becomes gradually higher than the theoretical value, and the relative error is increased, mainly due to the following reasons:

(1) The theoretical model is established based on related assumptions, elastic-plastic mechanics, a high-temperature stress relaxation model, and the material constitutive relationship, which contain errors to a certain extent.

(2) Since the thermal expansion properties of the material, the effects of boundary conditions, and springback itself cannot be ignored, the theoretical model established in this paper does not include the thermal expansion properties of the material, which increases the error. Furthermore, it also explains the reason why the experimental value deviates from the theoretical value as the amount of bending deformation increases.

## 5. Conclusions

In this study, uniaxial tension tests were carried out to obtain the true stress–strain model of material and stress relaxation experiments were carried out to develop the stress relaxation model. Based on both models, an analytical study of springback straightening after bending of the U-section of TC4 material under high-temperature conditions has been established. Finally, the analytical model was verified. The following conclusions have been derived based on the obtained results:The results of the uniaxial tension tests showed that the elastic-power hardening model can be used to establish the constitutive relationship of TC4 materials, which can more accurately describe the true stress–strain relationship of materials. Furthermore, the results of stress relaxation experiments showed that the stress relaxation equation describing natural logarithmic law is consistent with the typical stress relaxation law;Based on the theory of elastic-plastic mechanics, combined with the elastic-power hardening model and stress relaxation model, this paper established the springback and residual relative curvature equations at different bending degrees. Then, the springback and residual deflection equations were further derived. The results of the study showed that the dislocations and cavity structures generated by plastic deformation have high potential energy due to the lattice distortion caused by elastic deformation. This transformation will be strengthened with the increase of high-temperature time, and the elastic distortion will gradually become unrecovered creep distortion. Then, the corresponding elastic stress becomes smaller and smaller. Furthermore, the elastic strain part of the total strain is reduced and the plastic strain part is increased, so the stress is gradually reduced and the relative springback curvature is reduced;The results of the experiments showed that the relative errors between the theoretical residual deflections and residual deflections obtained by the tests are less than 20%, and the average absolute error is less than 10%. Therefore, the adopted hardening models can achieve an allowable relative error if hardening parameters are appropriately selected. By comparing and analyzing the error between the theoretical model and experiments, the theoretical, analytical model was established based on related assumptions, a high-temperature stress relaxation model, and the material constitutive model, which contained errors to a certain extent. Furthermore, the thermal expansion properties of the material, boundary conditions, and springback itself cannot be ignored.

## Figures and Tables

**Figure 1 materials-13-01895-f001:**
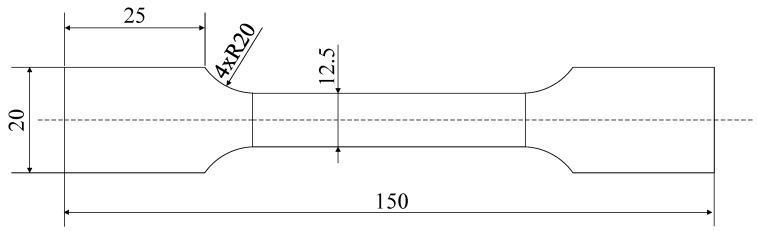
The geometry of the specimen used for the uniaxial tensile tests (unit: mm).

**Figure 2 materials-13-01895-f002:**
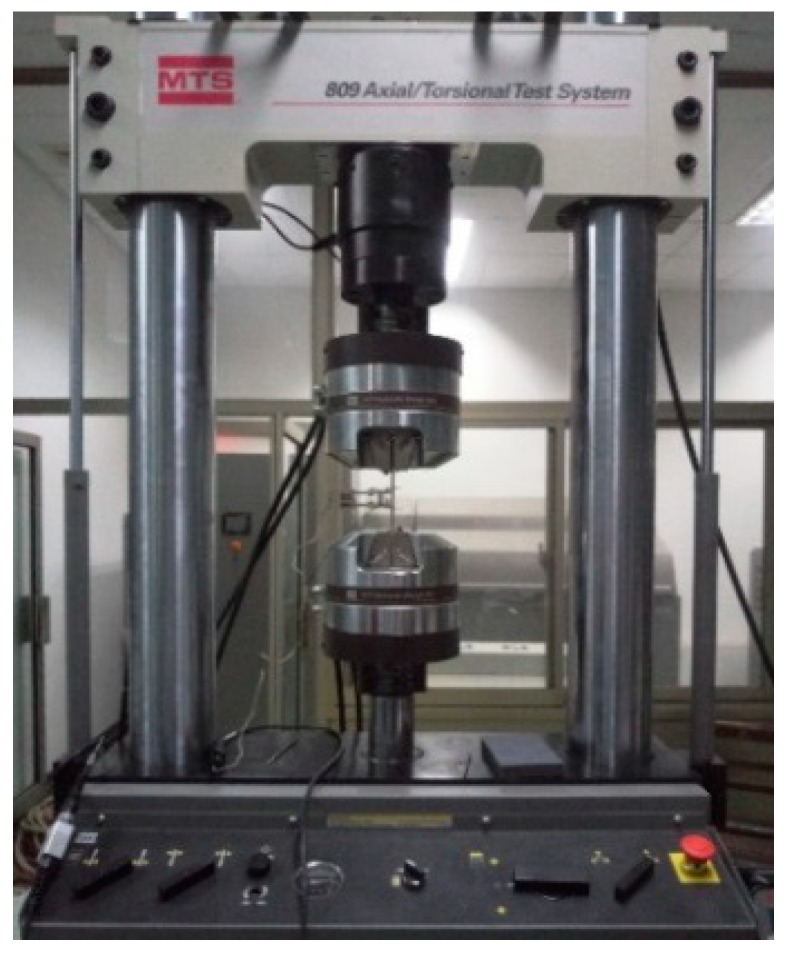
The electro-hydraulic universal testing machine MTS.

**Figure 3 materials-13-01895-f003:**
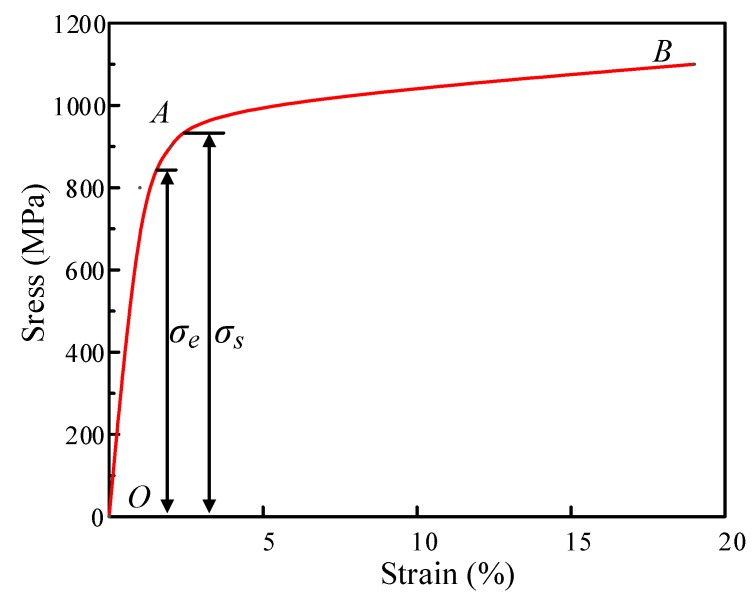
The true stress–strain curve of TC4.

**Figure 4 materials-13-01895-f004:**
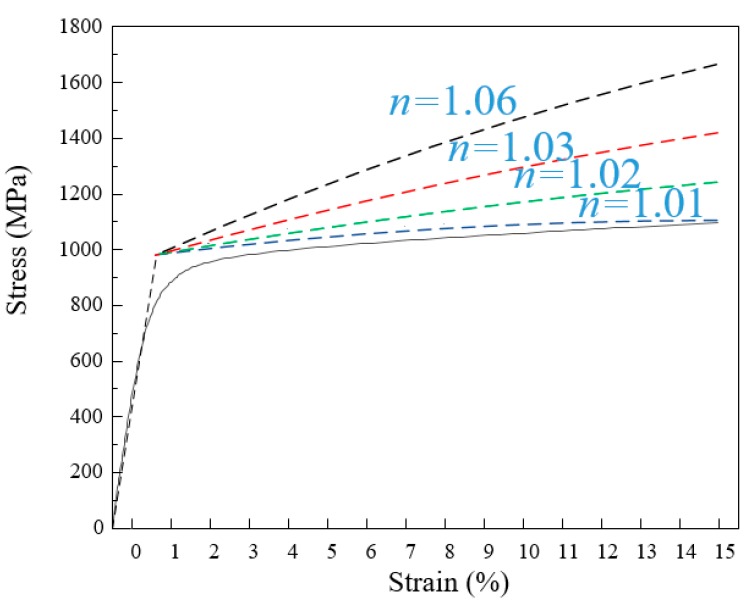
Comparison of stress–strain curves.

**Figure 5 materials-13-01895-f005:**
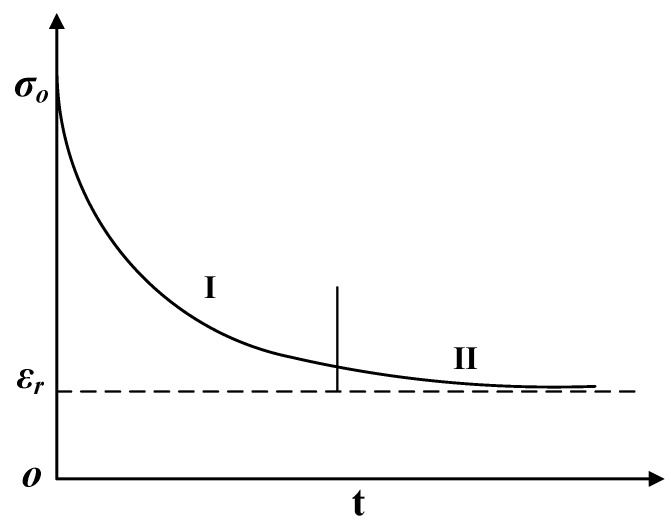
Typical stress relaxation curve.

**Figure 6 materials-13-01895-f006:**
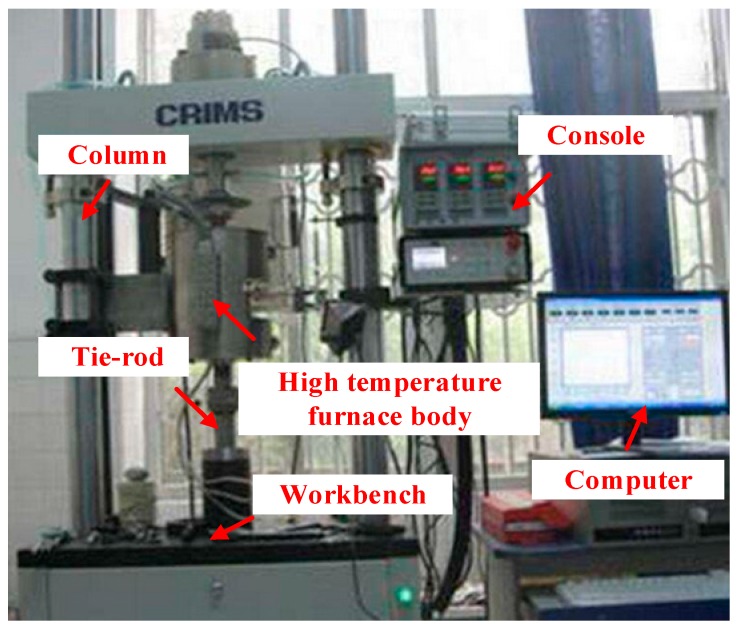
The high-temperature stress relaxation machine.

**Figure 7 materials-13-01895-f007:**
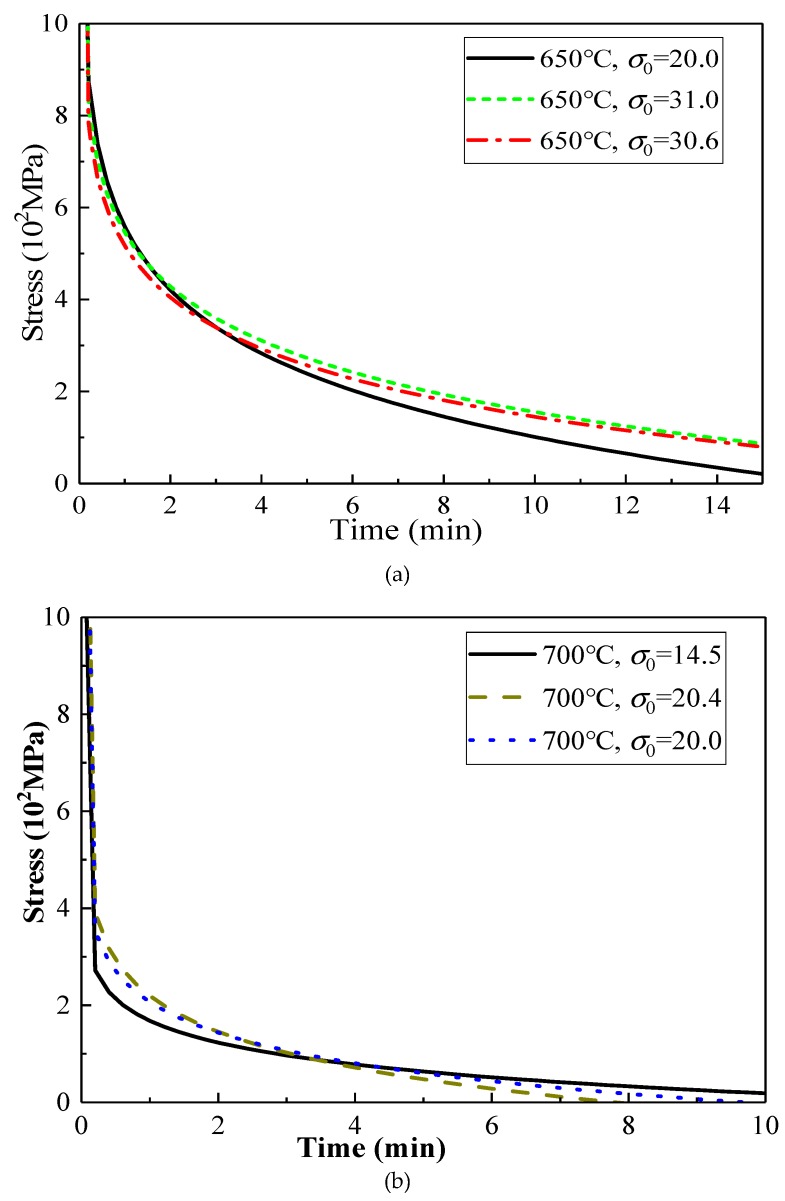
Stress relaxation equations under different temperatures and pressures. (**a**) The stress–strain curve at 650 °C; (**b**) the stress–strain curve at 700 °C; (**c**) stress–strain curve when $\sigma_{0}$ is about 20 MPa; (**d**) stress–strain curve when $\sigma_{0}$ is about 14 MPa.

**Figure 8 materials-13-01895-f008:**
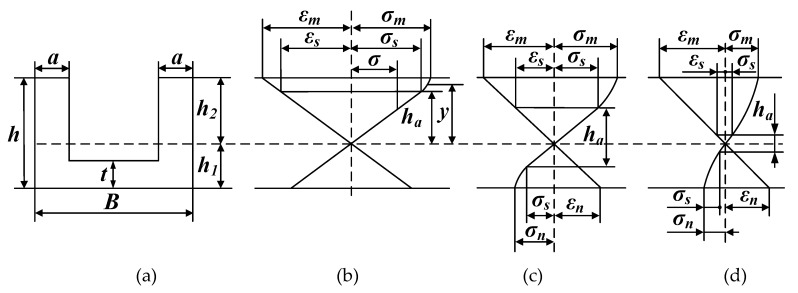
The U-section beam size and the stress–strain relationship under different bending degrees. (**a**) Geometry of the section; (**b**) first stage; (**c**) second stage; (**d**) third stage.

**Figure 9 materials-13-01895-f009:**
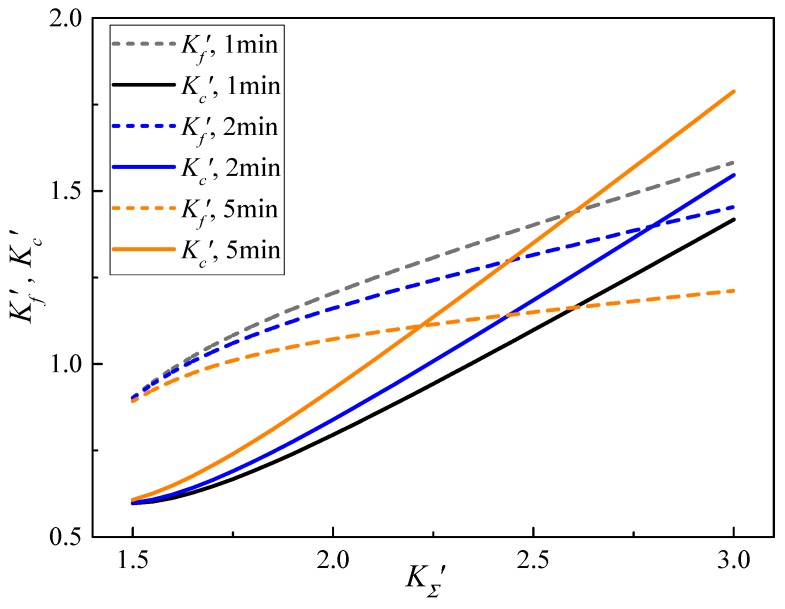
Springback and residual relative curvature equation curve.

**Figure 10 materials-13-01895-f010:**
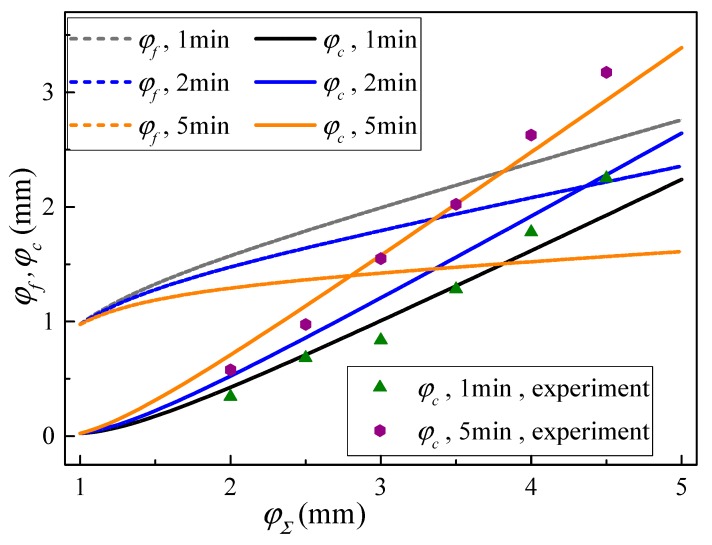
Springback and residual deflection curve.

**Table 1 materials-13-01895-t001:** Experimental material components [17].

Al, %	V, %	Fe, %	O, %	C, %	Ti, %
≤6.30	≤4.06	≤0.11	≤0.20	≤0. 01	Other

**Table 2 materials-13-01895-t002:** Stress relaxation equations under different temperatures.

Temperatures (°C)	$\sigma_{0}\;(\mathrm{MPa})$	Stress Relaxation Equations
650	20.0	σ = 20.0-1.984·ln(1+1.434E3·T)
650	31.0	*σ* = 31.0-1.695·ln(1+3.507E6·T)
650	30.6	σ = 30.6-1.612·ln(1+7.1379E6·T)
700	14.5	σ = 14.5-0.65·ln(1+3.6675E8·T)
700	20.4	σ = 20.4-1.07·ln(1+2.4431E7·T)
700	20.0	σ = 20.0-0.91·ln(1+3.618E8·T)
750	12.5	*σ* = 12.5-0.298·ln(1+9.8152E16·T)
750	13.0	*σ* = 13-0.424·ln(1+2.412E12·T)

**Table 3 materials-13-01895-t003:** Geometry of the U-section beam.

*a* (mm)	*B* (mm)	*h* (mm)	*h*_1_ (mm)	*h*_2_ (mm)	*t* (mm)	2*l* (mm)
2.5	115	70	18.74	51.26	3	500

**Table 4 materials-13-01895-t004:** Experimental results of springback.

Time(s)	Loading Stroke	Theoretical Model		Experiments		Relative Error
	$\varphi_{\sum }(\mathrm{mm})$	$\varphi_{f}(\mathrm{mm})$	$\varphi_{c}(\mathrm{mm})$	*φ_f_* (mm)	*φ_c_* (mm)	*φ_c_* (%)
60	2	1.574	0.426	1.756	0.344	19.25
60	2.5	1.790	0.710	1.817	0.683	3.80
60	3	1.993	1.007	2.163	0.837	16.88
60	3.5	2.189	1.311	2.218	1.282	2.21
60	4	2.382	1.618	2.220	1.780	−10.01
60	4.5	2.572	1.928	2.249	2.251	−16.75
300	2	1.292	0.708	1.422	0.578	18.36
300	2.5	1.364	1.136	1.526	0.974	14.26
300	3	1.422	1.578	1.451	1.549	1.84
300	3.5	1.474	2.026	1.477	2.023	0.15
300	4	1.522	2.478	1.373	2.627	−6.01
300	4.5	1.567	2.933	1.335	3.175	−8.25
	**Mean absolute error (%)**					**9.82**

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
