# Peer review of "Study on Springback Straightening after Bending of the U-Section of TC4 Material under High-Temperature Conditions"

_materials, 2020, doi:10.3390/ma13081895_

Round 1

Reviewer 1 Report

The manuscript deals with the springback effect after bending of thin-walled profile, which is an important problem from the engineering practice point of view.

In my opinion, some critical points need to be addressed and are listed below:

How many specimens were tested in tensile? Are E and limit stress values medium or based only on one specimen measurements?

Line 123: 0.2mm/min is the displacement rate - not the loading rate

Line 128: Curve on figure 3 is engineering stress-strain relation not true stress-strain

Line 153: Caption below Fig.4 should be more specific

Line 181: Provide more details about the experiment. Schematic of test stand or photo should be added.

Line 189: There is a statement that logarithmic equation can better describe stress relaxation . Please explain why? There is no comparison to other laws of stress relaxation.

Line 200: "... process from the surface to the inside." - I think that the word outer is missing.

Lines 203-210: This model is commonly known as Euler–Bernoulli beam and this name should be noted.

Line 240: K'c is used, but one line below in equation there is K'

Line 272: I think there is some mistake in translation. How you can compare together different physical values (deflection and bending load)? 

Line 288: What does it means that: "In this paper, the relationship between materials is established...". Maybe it is some translation error or you missed some words.

Line 302, 309: Caption should be more specific.

Line 299-301: Please provide some information about the experimental research: description, photographs or schematics of the test stand, devices to measurements, etc.

Line 316: Please explain why errors about 20% compared to experimental results are sufficient to prove the correctness of the adopted analytical model

Author Response

How many specimens were tested in tensile? Are E and limit stress values medium or based only on one specimen measurements?

Response 1: The uniaxial tensile tests of the unprocessed specimen of three were carried out on the electro-hydraulic universal testing machine MTS. The measurement data were deal with average, and then draw to curve according to these data.

Line 123: 0.2mm/min is the displacement rate - not the loading rate

Response 2: It has been modified to “select the displacement rate of 0.2mm/min”.

Line 128: Curve on figure 3 is engineering stress-strain relation not true stress-strain

Response 3: It has been modified in the paper.

Line 153: Caption below Fig.4 should be more specific

Response 4: it has been modified in the paper.

Line 181: Provide more details about the experiment. Schematic of test stand or photo should be added.

Response 5: It has been added some photos of experiment.

Line 189: There is a statement that logarithmic equation can better describe stress relaxation . Please explain why? There is no comparison to other laws of stress relaxation.

Response 6: the sentence is mistake, it has been modified to “It can be seen from Figure 7 that the stress relaxation law of TC4 material at high temperature shows consistency with the typical stress relaxation law of Figure 5.”.

Line 200: "... process from the surface to the inside." - I think that the word outer is missing.

Response 7: It has been modified to “…process from the outer surface to inside”.

Lines 203-210: This model is commonly known as Euler–Bernoulli beam and this name should be noted.

Response 8: It has been modified to “…follows the Euler–Bernoulli beam theory”.

Line 240: K'c is used, but one line below in equation there is K'

Response 9: It has been modified in the paper.

Line 272: I think there is some mistake in translation. How you can compare together different physical values (deflection and bending load)? 

Response 10: It has been modified “bending load” to “loading stroke”.

Line 288: What does it means that: "In this paper, the relationship between materials is established...". Maybe it is some translation error or you missed some words.

Response 11: It has been modified “the constitutive relationship of the material is established based on the elastic-power hardening model”.

Line 302, 309: Caption should be more specific.

Response 12: Captions have been modified to “Experimental results of springback” and “Springback and residual deflection curve”.

Line 299-301: Please provide some information about the experimental research: description, photographs or schematics of the test stand, devices to measurements, etc.

Response 13: It has been modified in the paper.

Line 316: Please explain why errors about 20% compared to experimental results are sufficient to prove the correctness of the adopted analytical model

Response 14: It has been modified to “Thus, although there is an apparent yield stage in the true stress-strain curve, the adopted hardening models can achieve an allowable relative error, if hardening parameters are properly selected”.

Reviewer 2 Report

I must admit I am impressed with the article. It may not deal with new and unexplored issues, but the comprehensive approach to the topic, the analyzes performed and the transparent methodology deserve recognition. The introduction to the topic is transparent, the purpose of the thesis is clearly defined and the subject is shown in the context of other scientific papers. The methodology and description of the material used is also unquestionable, as is the course of the research. However, I miss one element - namely showing the structure of the material after a series of tests - although - if such tests are carried out and supplemented, they can constitute material for a separate article. The conclusions are consistent, reflect the sense of the work.

Author Response

Thank you for your comments.

Reviewer 3 Report

The subject of the article concerns the assessment of springback effect of U-section samples made of titanium alloys at high temperatures. While the subject seems to be very interesting, the presented work requires corrections, in particular in the introduction, where more detailed presentation of the considered problem is needed. In the experimental part, apart the purely mechanical approach, there is no approach taking into account material tests, which are key factor in profile of journal "Materials" with so high IF. There is also no detailed discussion of obtained results especially in field of comparison with results of other authors. Below are some comments:
1. The introduction should explain the springback effect, where and when it occurs, and what are its consequences.

2. Literature data and detailed arguments which explain the need for research in this direction and data illustrating the scale of the problem were not given.

3. The purpose of the article is not clearly explained in the introduction.

4. There is a lack of structural research in the experimental part, even ordinary microscopic observations - biphasic titanium alloys under high temperatures and under high load conditions undergo many structural changes.  Therefore this problem requires more comprehensive approach, especially when was send to the journal with "material" profile.

5. The value and unit should be written separately, i.e. lines:124,135, etc.

Author Response

  1. The introduction should explain the springback effect, where and when it occurs, and what are its consequences.

Response 1: It has been modified in the paper

  1. Literature data and detailed arguments which explain the need for research in this direction and data illustrating the scale of the problem were not given.

Response 2: It has been modified in the paper

.

  1. The purpose of the article is not clearly explained in the introduction.

Response 3: It has been modified in the paper

  1. There is a lack of structural research in the experimental part, even ordinary microscopic observations - biphasic titanium alloys under high temperatures and under high load conditions undergo many structural changes.  Therefore, this problem requires more comprehensive approach, especially when was send to the journal with "material" profile.

Response 4: It has been add more details experimental part.

  1. The value and unit should be written separately, i.e. lines:124,135, etc.

Response 5:  It has been modified in the paper.

Reviewer 4 Report

Thank you for providing the manuscript "Evaluation of springback after bending of U-section of TC4 material under high-temperature conditions"
Below are comments, please correct and send comments.
How did you prepare tensile test specimens?
What caused the uneven voltage jumps on the curve in figure three? Please explain or change.
The quality of the prescribed formulas in section three is poor. Please change.

Author Response

How did you prepare tensile test specimens?

Response 1: According to the tensile test standard, the tensile test standard parts are first produced by lathe, and then the tensile test samples of corresponding size are produced by WEDM.

What caused the uneven voltage jumps on the curve in figure three? Please explain or change.

Response 2: When the stress is lower than , the stress becomes proportional to the strain of the sample, then the stress is removed and the deformation disappears. It indicates that the sample is in the elastic deformation stage. When the stress exceeds , the linear relationship between stress and strain is broken. If unloading, the deformation of the sample can only be partially restored, while some residual deformation is retained, that is, plastic deformation, which indicates that the deformation of the material has entered the elastic-plastic deformation stage.

The quality of the prescribed formulas in section three is poor. Please change.

Response 3: It has been modified in the paper.

did you prepare tensile test specimens?

Response 1: According to the tensile test standard, the tensile test standard parts are first produced by lathe, and then the tensile test samples of corresponding size are produced by WEDM.

What caused the uneven voltage jumps on the curve in figure three? Please explain or change.

Response 2: When the stress is lower than , the stress becomes proportional to the strain of the sample, then the stress is removed and the deformation disappears. It indicates that the sample is in the elastic deformation stage. When the stress exceeds , the linear relationship between stress and strain is broken. If unloading, the deformation of the sample can only be partially restored, while some residual deformation is retained, that is, plastic deformation, which indicates that the deformation of the material has entered the elastic-plastic deformation stage.

The quality of the prescribed formulas in section three is poor. Please change.

Response 3: It has been modified in the paper.

Round 2

Reviewer 1 Report

Authors addressed my comments. I recommend the publication of this work.

I suggest only correcting two graphic problems:

1. Annotations in Figure 6 are hardly visible. It would be good to consider changing the text colour or adding a white background.

2. In Fig. 7 there are errors in the symbol of Celsius in the legend.

Author Response

  1. Annotations in Figure 6 are hardly visible. It would be good to consider changing the text colour or adding a white background.

Response 1: It has been modified in the paper.

  1. In Fig. 7 there are errors in the symbol of Celsius in the legend.

Response 2: It has been modified in the paper.

Reviewer 3 Report

All my comments were improved. English spell checked is required.

Author Response

Response: It has been modified in the paper.